# Robust Aggregation for Federated Learning by Minimum *γ*-Divergence Estimation

**DOI:** 10.3390/e24050686

**Published:** 2022-05-13

**Authors:** Cen-Jhih Li, Pin-Han Huang, Yi-Ting Ma, Hung Hung, Su-Yun Huang

**Affiliations:** 1Institute of Statistical Science, Academia Sinica, Taipei City 11529, Taiwan; licz82@stat.sinica.edu.tw (C.-J.L.); erics910085@gmail.com (Y.-T.M.); 2Data Science Degree Program, National Taiwan University, Taipei City 10617, Taiwan; peterkonerko@gmail.com; 3Institute of Epidemiology and Preventive Medicine, National Taiwan University, Taipei City 10055, Taiwan; hhung@ntu.edu.tw

**Keywords:** byzantine problem, density power divergence, federated learning, γ-divergence, influence function, robustness

## Abstract

Federated learning is a framework for multiple devices or institutions, called local clients, to collaboratively train a global model without sharing their data. For federated learning with a central server, an aggregation algorithm integrates model information sent from local clients to update the parameters for a global model. Sample mean is the simplest and most commonly used aggregation method. However, it is not robust for data with outliers or under the Byzantine problem, where Byzantine clients send malicious messages to interfere with the learning process. Some robust aggregation methods were introduced in literature including marginal median, geometric median and trimmed-mean. In this article, we propose an alternative robust aggregation method, named γ-mean, which is the minimum divergence estimation based on a robust density power divergence. This γ-mean aggregation mitigates the influence of Byzantine clients by assigning fewer weights. This weighting scheme is data-driven and controlled by the γ value. Robustness from the viewpoint of the influence function is discussed and some numerical results are presented.

## 1. Introduction

Federated learning (FL), a distributed learning paradigm, was proposed by Google in 2016 [1] for distributing the optimization of model parameters over multiple machines. The distributed framework allows the data to be stored in local devices without sharing with each other. These machines, called clients, collaboratively train a global model. In this article, we consider the case that there is a central server, which works as a coordinator and aggregates the gradient information sent by the local clients to update the global model through an iterative process [2]. In real-world applications, clients, such as hospitals and clinical institutions, hold highly sensitive and private data such as electronic healthcare records, medical images, etc. [3]. However, FL still encounters several challenges in a real world setting. In its training process, some unreliable clients may produce outlying data information or even send malicious values [4], which leads to a biased result or even a failure of the training process. These arbitrary clients are called Byzantine clients. With a properly designed aggregation scheme, the effect by Byzantine clients can be reduced or excluded. Several papers have made significant contributions to developing Byzantine-resilience methods with specific use of robust aggregators, including marginal median [5], geometric median [6], and trimmed-mean [7]. Another aggregation framework is the Byzantine-robust stochastic aggregation method [8], which adds a regularization term in the objective function to handle the Byzantine and heterogeneous data problem.

In this article, we consider the case of FL, which consists of a central server and *m* clients. Assume that there is an α-fraction of Byzantine clients. Our goal is to minimize the following objective function:(1)$$L\left(\theta \right)=\sum_{i=1}^{m}\pi_{i}E_{\eta_{i}∼D}\left[l_{\eta_{i}}\left(\theta \right)\right],$$
where $\theta \in R^{p}$ is the global model parameter, $l_{\eta_{i}}\left(\theta \right)$ is the loss function of client *i* with local data $\eta_{i}$ having an unknown distribution D, and ${\left\{\pi_{i}\right\}}_{i=1}^{m}$ are the prior weights assigned to local clients. At the tth round of FL iterative updates, the server broadcasts the current parameter value $\theta^{t}\in R^{p}$ to local clients. Normal clients faithfully compute an estimate of the local gradients $E_{\eta_{i}∼D}\nabla l_{\eta_{i}}\left(\theta^{t}\right)$ and send the gradient information back to the server. On the contrary, Byzantine clients send arbitrary erroneous messages to obstruct the optimization process. The central server aggregates the gradient information from clients and updates the model parameter from $\theta^{t}$ to $\theta^{t+1}$. At the end, the server should output an estimate of the optimal θ, which aims to minimize the objective loss (Equation 1). In this work we consider uniform client weights, i.e., $\pi_{1}=\cdots =\pi_{m}=1$. The main reason is that, under the Byzantine problem, the local sample sizes claimed by Byzantine clients might not be reliable.

Our main contributions of this work are listed below.

We propose the γ-mean as a robust aggregator in federated learning. This robust aggregator mitigates the influence of Byzantine clients by assigning fewer weights. The weighting scheme is data-driven and controlled by the tuning parameter γ.We have a discussion on robustness from the influence function point of view. Benefits of adopting γ-mean can be seen from its influence function in comparison to other robust alternatives such as marginal median, geometric median and trimmed mean.The robustness of γ-mean is then verified through simulation study and real data experiments. The γ-mean based aggregation outperforms other robust aggregators such as marginal median, geometric median and trimmed mean.

The rest of the article is organized as follows. In Section 2, we review some related work in robust federated learning. In Section 3, we propose our main method for robust FL aggregation and provide some theoretical viewpoints by the influence function. In Section 4, we conduct an extensive simulation study. In Section 5, we provide some further numerical examples using MNIST, fashion MNIST and chest X-ray images (pneumonia). In Section 6, we make some concluding remarks.

## 2. Related Work

McMahan et al. introduced the *FedAvg* algorithm [9], which used the mean as its aggregator. The sample mean aggregation is a very popular FL framework [9,10]. However, the sample mean is known to be vulnerable to outliers and heavy-tailed distribution. For instance, Byzantine clients [7,8,11] may send extreme values to strongly influence the aggregation of sample mean. There are some robust alternatives to the mean aggregator, such as marginal median, geometric median and trimmed mean. In contrast to mean, marginal median is a relatively robust aggregator by computing coordinate-wise medians. Another Byzantine-resilient aggregator is geometric median, which computes the geometric median of ${\left\{x_{i}\right\}}_{i=1}^{m}$ given by (see Weiszfeld’s algorithm [12] for solution computation).
(2)$$\mathrm{GeoMed}\left({\left\{x_{i}\right\}}_{i=1}^{m}\right)=\arg\min_{x\in R^{p}}\sum_{i=1}^{m}{∥x-x_{i}∥}_{2}.$$

The trimmed mean aggregator computes coordinate-wise trimmed means by removing the β-fraction of data points from each of the two tails [11]. Both the marginal median and geometric median, though are robust to outliers, but are not efficient, as they only use the most centrally-lying data for inference. As for trimmed mean, first it needs a proper selection of β. Second, it often does not work well for outliers lying on one side of a coordinate instead of on two sides. To have a better balance between robustness and efficiency, we will propose another robust aggregator based on a robust density power divergence, namely, the γ-divergence.

## 3. Proposed Aggregator and Its Robustness

In this section, we introduce an aggregator, called “γ-mean”, which is based on the minimum γ-divergence estimation. Two versions of implementation algorithms are given and some robustness viewpoints based on the influence function are provided.

### 3.1. Minimum γ-Divergence Estimation

The γ-divergence [13,14], which is also known as the density power divergence of type zero [15], is a robust divergence against outliers with a tuning parameter γ>0. The γ-divergence between the data distribution with probability density function *g* and the model distribution $f_{\tau }$ (indexed by τ) is defined by
$$D_{\gamma }\left(g,f_{\tau }\right)=\frac{{∥g∥}_{\gamma +1}}{\gamma (\gamma +1)}\left\{1-\int {\left(\frac{f_{\tau }\left(x\right)}{{∥f_{\tau }∥}_{\gamma +1}}\right)}^{\gamma }\frac{g(x)}{{∥g∥}_{\gamma +1}}dx\right\},$$
where ${∥f_{\tau }∥}_{\gamma +1}={\left(\int f_{\tau }^{\gamma +1}\left(x\right)dx\right)}^{1/(\gamma +1)}$. In the limiting case, it reduces to the Kullback-Leibler divergence, i.e., $\lim_{\gamma \to 0}D_{\gamma }\left(g,f_{\tau }\right)=\int \ln\left(\frac{g(x)}{f_{\tau }\left(x\right)}\right)g\left(x\right)dx$. In the population level, the minimum γ-divergence estimation of τ is given by
(3)$$\tau_{\gamma }^{*}=\arg\min_{\tau }D_{\gamma }\left(g,f_{\tau }\right)=\arg\max_{\tau }\int {\left(\frac{f_{\tau }\left(x\right)}{{∥f_{\tau }∥}_{\gamma +1}}\right)}^{\gamma }g\left(x\right)dx.$$

In the sample level with empirical data ${\left\{x_{i}\right\}}_{i=1}^{m}$, the data distribution will be replaced by the empirical distribution to get the estimate ${\hat{\tau }}_{\gamma }$:$${\hat{\tau }}_{\gamma }=\arg\min_{\tau }D_{\gamma }\left(g,f_{\tau }\right)=\arg\max_{\tau }\frac{1}{m}\sum_{i=1}^{m}{\left(\frac{f_{\tau }\left(x_{i}\right)}{{∥f_{\tau }∥}_{\gamma +1}}\right)}^{\gamma }.$$

In this article, we use a Gaussian working model, i.e., $f_{\tau }∼N\left(\mu ,\Sigma \right)$. Suppose *g* takes a contaminated form: $g\left(x\right)=\left(1-\epsilon \right)f_{\tau }\left(x\right)+\epsilon h\left(x\right)$, where h(x) is the probability density function of contamination distribution. It is assumed that $\int f_{\tau }^{\gamma }\left(x\right)h\left(x\right)$ is small for certain γ>0, so that the contamination has little effect in the learning process. By taking derivatives with respect to τ=(μ,Σ) and setting them to zero, the estimates $\left(\hat{\mu },\hat{\Sigma }\right)$ have to satisfy the following estimating equations:$$\begin{array}{c} \frac{1}{m}\sum_{i=1}^{m}d_{i}^{\left(\gamma \right)}\left(\hat{\mu },\hat{\Sigma }\right)\left(x_{i}-\hat{\mu }\right)=0,\\ \frac{1}{m}\sum_{i=1}^{m}d_{i}^{\left(\gamma \right)}\left(\hat{\mu },\hat{\Sigma }\right)\left\{\frac{1}{\gamma +1}\hat{\Sigma }-\left(x_{i}-\hat{\mu }\right){\left(x_{i}-\hat{\mu }\right)}^{⊤}\right\}=0, \end{array}$$
where $d_{i}^{\left(\gamma \right)}\left(\hat{\mu },\hat{\Sigma }\right)=\exp\left(\frac{-\gamma }{2}{\left(x_{i}-\hat{\mu }\right)}^{⊤}{\hat{\Sigma }}^{-1}\left(x_{i}-\hat{\mu }\right)\right)$. The solution pair, $\left(\hat{\mu },\hat{\Sigma }\right)$, has to satisfy the following stationary equations: (4)$$\begin{array}{ccc} \hat{\mu } & = & \frac{\sum_{i=1}^{m}d_{i}^{\left(\gamma \right)}\left(\hat{\mu },\hat{\Sigma }\right)x_{i}}{\sum_{i=1}^{m}d_{i}^{\left(\gamma \right)}\left(\hat{\mu },\hat{\Sigma }\right)}, \end{array}$$(5)$$\begin{array}{ccc} \hat{\Sigma } & = & \left(\gamma +1\right)\;\frac{\sum_{i=1}^{m}d_{i}^{\left(\gamma \right)}\left(\hat{\mu },\hat{\Sigma }\right)\left(x_{i}-\hat{\mu }\right){\left(x_{i}-\hat{\mu }\right)}^{⊤}}{\sum_{i=1}^{m}d_{i}^{\left(\gamma \right)}\left(\hat{\mu },\hat{\Sigma }\right)}. \end{array}$$

### 3.2. Robust Aggregation by γ-Mean

In view of the stationary Equations (Equation 4) and (5), we use a fixed-point iteration. Two algorithms are provided below. Algorithm 1 is the usual γ-mean, which uses N(μ,Σ) as the working model, and Algorithm 2 is the simple γ-mean, which adopts $N(\mu ,\sigma^{2}I)$ as the working model. Note that, in the former case when the sample size is not sufficiently large enough to have a stable estimate of the covariance inverse, we will use ${\left\{\mathrm{diag}\left(\hat{\Sigma }\right)\right\}}^{-1}$ instead, where $\hat{\Sigma }$ is the minimum γ-divergence estimator for the covariance and $\mathrm{diag}(\hat{\Sigma })$ denotes the diagonal matrix consisting of diagonal elements of $\hat{\Sigma }$. Also note that, in the latter case of simple γ-mean, $\sigma^{2}$ can be merged into the hyperparameter γ. Thus, we will simply use the standard Gaussian as the working model.
**Algorithm 1** γ-mean with a Gaussian working model.**Input:** Gradient information $X=[x_{1},x_{2},$…$x_{m}]$ and the maximum number of iterations *S***Output:**$\mu_{\gamma }$ = $\hat{\mu }$  Start with initials $\hat{\mu }={\hat{\mu }}_{ini}$ and $\hat{\Sigma }={\hat{\Sigma }}_{ini}$.  **for** s=1,2,…,S (while s≤S and iterations not yet converge) **do**     **for** i=1,2,…,m **do**       Calculate $d_{i}^{\left(\gamma \right)}\left(\hat{\mu },\hat{\Sigma }\right)\leftarrow \exp\left(-\frac{\gamma }{2}{\left(x_{i}-\hat{\mu }\right)}^{⊤}{\hat{\Sigma }}^{-1}\left(x_{i}-\hat{\mu }\right)\right)$ at the *i*th local client.     **end for**     Denote $w_{i}^{\left(\gamma \right)}=\frac{d_{i}^{\left(\gamma \right)}\left(\hat{\mu },\hat{\Sigma }\right)}{\sum_{i=1}^{m}d_{i}^{\left(\gamma \right)}\left(\hat{\mu },\hat{\Sigma }\right)}$.     $\hat{\mu }\leftarrow \sum_{i=1}^{m}w_{i}^{\left(\gamma \right)}\cdot x_{i}$,     $\hat{\Sigma }\leftarrow \left(1+\gamma \right)\cdot \sum_{i=1}^{m}w_{i}^{\left(\gamma \right)}\cdot \left(x_{i}-\hat{\mu }\right){{(x}_{i}-\hat{\mu })}^{⊤}$.**end for**

**Algorithm 2** simple γ-mean with the standard Gaussian as the working model.
**Input:** Gradient information $X=[x_{1},x_{2},$…$x_{m}]$ and the maximum number of iterations *S*
**Output:**$\mu_{\gamma }$ = $\hat{\mu }$
  Start with initial $\hat{\mu }={\hat{\mu }}_{ini}$.
  **for** s=1,2,…,S (while s≤S and iterations not yet converge) **do**
     **for** i=1,2,…,m **do**
       Calculate $d_{i}^{\left(\gamma \right)}\left(\hat{\mu }\right)\leftarrow \exp\left(-\frac{\gamma }{2}{\left(x_{i}-\hat{\mu }\right)}^{⊤}\left(x_{i}-\hat{\mu }\right)\right)$ at the ith local client.
     **end for**
     Denote $w_{i}^{\left(\gamma \right)}=\frac{d_{i}^{\left(\gamma \right)}\left(\hat{\mu }\right)}{\sum_{i=1}^{m}d_{i}^{\left(\gamma \right)}\left(\hat{\mu }\right)}$.
     $\hat{\mu }\leftarrow \sum_{i=1}^{m}w_{i}^{\left(\gamma \right)}\cdot x_{i}$.

**end for**



For the extremely large dimension *p*, such as in a deep neural network model, the simple γ-mean algorithm will be more feasible than the usual γ-mean for better numerical stability.

### 3.3. Robustness

#### 3.3.1. Influence Function

The influence function is a tool to evaluate the change of an estimator by a small perturbation to the data distribution. An estimator with a lower influence provides better resistance against outliers. Therefore, we analyze the robustness of different estimators by showing their influence functions. Let ${\left\{x_{i}\right\}}_{i=1}^{m}\subset R^{p}$ be sampled from *G* and *T* be the statistical functional for estimation. The robustness of T(G) can be evaluated by the influence function of *T*:$${\mathrm{IF}}_{T}\left(x_{0};G\right)={\left.\frac{\partial }{\partial \epsilon }T\left\{\left(1-\epsilon \right)G+\epsilon \delta_{x_{0}}\right\}\right|}_{\epsilon =0},$$
where $\delta_{x_{0}}$ is the point mass at $x_{0}$. In other words, the influence function of *T* is the Gâteaux derivative of T(G) with respect to *G* along the direction $\delta_{x_{0}}-G$. In this paper, we consider evaluating the influence function at the Gaussian working model $F_{\tau }$ with τ=(μ,Σ) and seeing the deviant effect when a point perturbation $x_{0}$ is added. The influence function of M-estimators have been well-studied in [16], which is related to the inverse of Hessian matrix and the first order derivative. The estimator based on γ-divergence is also an M-estimate. Its influence function derivation can be found in [14] and is given by
(6)$${\mathrm{IF}}_{\mu_{\gamma }}\left(x_{0};G\right){|}_{G=F_{\tau }}={\left(\gamma +1\right)}^{\frac{p+2}{2}}\exp\left(\frac{-\gamma }{2}{\left(x_{0}-\mu_{\gamma }\right)}^{⊤}\Sigma_{\gamma }^{-1}\left(x_{0}-\mu_{\gamma }\right)\right)\left(x_{0}-\mu_{\gamma }\right),$$
where $\mu_{\gamma }$ and $\Sigma_{\gamma }^{-1}$ are the minimum γ-divergence estimators for the location and covariance matrix of the working model $F_{\tau }$, respectively. The influence function of the γ-mean, ${\mathrm{IF}}_{\mu_{\gamma }}\left(x_{0};F_{\tau }\right)$, is similar to the influence function of the sample mean, which is given by $\left(x_{0}-\mu \right)$, except for an additional multiplicative weight $d_{x_{0}}^{\left(\gamma \right)}\left(\mu_{\gamma },\Sigma_{\gamma }\right)$. This weight focuses on the Mahalanobis distance between the perturbing point $x_{0}$ and $\mu_{\gamma }$. As a result, when $x_{0}$ is an outlier away from the target mean of data distribution, this weighting factor produces a down-weighting effect and prevents the estimator from biased estimation caused by the outlier.

#### 3.3.2. Comparison with Other Aggregators

In this subsection, we present influence functions for several robust aggregation methods including marginal median, geometric median and trimmed-mean.

Geometric median $\mu_{\mathrm{geo}}$: The influence function of $\mu_{\mathrm{geo}}$ is given by [17]
$$\begin{array}{ccc} {\mathrm{IF}}_{\mu_{\mathrm{geo}}}\left(x_{0};G\right) & = & {\left(E_{X}\left[{∥X-\mu_{\mathrm{geo}}∥}_{2}^{-1}\left\{I_{p}-{∥X-\mu_{\mathrm{geo}}∥}_{2}^{-2}\left(X-\mu_{\mathrm{geo}}\right){\left(X-\mu_{\mathrm{geo}}\right)}^{⊤}\right\}\right]\right)}^{-1}\\ & & \frac{x_{0}-\mu_{\mathrm{geo}}}{∥x_{0}-\mu_{\mathrm{geo}}{∥}_{2}}, \end{array}$$
where the expectation $E_{X}$ is taken with respect to the data distribution *G*.

Trimmed-mean $\mu_{\mathrm{trm}}$: The trimmed-mean is an *L*-esimator, and its influence function is derived in [18]. For the influence function of the marginal trimmed-mean, we derive it coordinate-wise since the coordinates are independent to each other. Therefore, let $G_{i}^{-1}$ be the quantile function of marginal data distribution on *i*-th component, and the influence function of $\mu_{\mathrm{trm}}$ is given by
$${\mathrm{IF}}_{\mu_{\mathrm{trm}}}\left(x_{0};G\right)=\frac{1}{1-2\beta }\left(\begin{array}{c} w_{1}\\ \vdots \\ w_{p} \end{array}\right),$$
where
$$w_{i}=\left\{\begin{array}{cc} G_{i}^{-1}\left(\beta \right)-W_{i}, & x_{0i}<G_{i}^{-1}\left(\beta \right),\\ x_{0}-W_{i}, & G_{i}^{-1}\left(\beta \right)<x_{0i}<G_{i}^{-1}\left(1-\beta \right),\\ G_{i}^{-1}\left(1-\beta \right)-W_{i}, & x_{0i}>G_{i}^{-1}\left(1-\beta \right), \end{array}\right.$$

$x_{0}={\left(x_{01},\cdots ,x_{0p}\right)}^{⊤}$, and
$$W_{i}=\left(1-2\beta \right)\int_{G_{i}^{-1}\left(\beta \right)}^{G_{i}^{-1}\left(1-\beta \right)}t\;dG_{i}\left(t\right)+\beta \left(G_{i}^{-1}\left(1-\beta \right)+G_{i}^{-1}\left(\beta \right)\right).$$

It is obvious that an influential effect, $x_{0}-W_{i}$, still remains if $x_{0}$ is outside the trimmed range.

Marginal median $\mu^{\left(0.5\right)}$: The influence function of the median is also derived in [18], and we use the same techniques as used in the trimmed-mean to derive the influence function of the marginal median. The influence function is given by
(7)$${\mathrm{IF}}_{\mu^{\left(0.5\right)}}\left(x_{0};G\right)=\left(\begin{array}{c} \frac{x_{01}-\mu_{1}^{\left(0.5\right)}}{2g_{1}\left(\mu_{1}^{\left(0.5\right)}\right)\left|x_{01}-\mu_{1}^{\left(0.5\right)}\right|}\\ \vdots \\ \frac{x_{0p}-\mu_{p}^{\left(0.5\right)}}{2g_{p}\left(\mu_{p}^{\left(0.5\right)}\right)\left|x_{0p}-\mu_{p}^{\left(0.5\right)}\right|} \end{array}\right),$$
where $\mu^{\left(0.5\right)}={\left(\mu_{1}^{\left(0.5\right)},\cdots ,\mu_{p}^{\left(0.5\right)}\right)}^{⊤}$ is the marginal median and $g_{i}$ is the density function of the marginal distribution $G_{i}$.

The influence functions given above reveal that different robust aggregators have different resistance ability against outliers. The trimmed mean can still be influenced by outliers outside the trimmed range. While the marginal median and the geometric median are fairly robust, they may lose too much efficiency due to the use of only the most centrally-lying data point. The γ-mean provides an adjustable control between the robustness and efficiency by the hyper-parameter γ.

## 4. Simulation Study

In this section, we evaluate our robust γ-mean aggregator and compare it with existing aggregators, including mean, marginal median, geometric median and trimmed mean, through simulation. For computing the geometric median, Weiszfeld’s algorithm [12] is used. In this simulation study, we investigate the behavior and inspect the robustness of the γ-mean and other aggregators as an estimator of the location parameter under multivariate Gaussian distribution and multivariate *t*-distribution with and without Byzantine interference. We build several simulation scenarios, including testing aggregators across the growth of dimension *p*, different fractions (i.e., α values) of Byzantine attacks and for various settings of γ (which controls the degree of robustness). All our simulation experiments were run on an Nvidia DGX A100 server. We used one A100 GPU card, 16G RAM, and 16 cores of AMD-EPYC-7742 CPUs. However, a personal computer with a moderate performance-efficient GPU card and CPU can also carry out the simulation experiments, but with longer run time. For the real data examples in Section 5, a personal computer will not be sufficient for carrying out the computing job.

### 4.1. Simulation Settings

Scenario 1. We focus on the behavior of aggregators for increasing *p* from 20 to 1000. Other experimental setting is as follows: the number of clients m=200, the fraction of Byzantine attacks α=0 and α=0.1, and the hyper-parameter for controlling the robustness γ=2/p.Scenario 2. We focus on the behavior of aggregators for increasing contamination fraction α from 0 to 0.5. Other experimental setting is as follows: m=200, p=1000 and γ=2/p.Scenario 3. We focus on the effect of γ values and set γ=c/p for various constants *c* ranging from 0.5 to 4. Other experimental setting is as follows: m=200, α=0.1 and *p* ranges from 1 to 1000.Scenario 4. After comparison between the γ-mean and other aggregators, we focus on the comparison between two versions of our proposal, the γ-mean versus the simple γ-mean. Other experimental setting is as follows: m=200, α=0.1, γ=2/p and *p* ranges from 1 to 1000.

For regular clients, gradient vectors are generated from the standard Gaussian distribution and *t*-distribution with 5 degrees of freedom. For Byzantine clients, gradient vectors are generated from the same Gaussian and *t* distributions, except with a location shift $\mu =100\times 1_{p}$, where $1_{p}$ is a vector with one in all entries. Each experimental scenario is implemented with 100 replicate runs.

### 4.2. Results

To compare the performance of different aggregators, the mean squared error (MSE) to the true target value is used as a performance metric. MSE can be further decomposed into the squared bias and variance.

#### 4.2.1. Scenario 1

The results are shown in Figure 1. Without contamination (Figure 1a), all aggregators have MSE close to zero. For the Gaussian case, the MSE curves of the mean, geometric median and γ-mean are almost collapsed together. These 3 curves have the lowest MSE values followed by the trimmed-mean and marginal median. For the *t*-distribution, the γ-mean has the lowest MSE, followed by the geometric median, trimmed-mean, mean and then marginal median. With 10% of contamination, the mean-aggregator fails to perform well and has large values of MSE, verifying that it is not a robust aggregator. Due its large MSE value, the result of the mean-aggregator is removed from Figure 1b. From Figure 1b we can see that γ-mean performs the best among the five aggregators in terms of MSE. In addition, γ-mean remains low and stable in variance under both Gaussian and *t* distributions.

#### 4.2.2. Scenario 2

As Figure 2 shows, values of squared bias are close to zero when α=0 and increase as α grows for all aggregators. However, the increasing velocity of aggregators are quite different. Mean-aggregator surges with the highest velocity, while γ-mean progresses with the lowest velocity. Trimmed-mean slowly grows when α≤0.1; yet, its increasing pace is about the same as the mean after α>0.1 due to the fixed 10% trimming percentage from both tails. The trends in MSE for different aggregators have similar patterns, and the only difference is in the scale with the γ-mean having the lowest MSE values. Also note that the variance of the marginal median increases much faster than other aggregators as α increases, followed by the geometric median, γ-mean, trimmed mean and then the mean. However, the trimmed mean and the mean have dominant bias leading to high MSE, even though they have small variance. As the values of squared bias are large for the mean and trimmed-mean, the bias curves for the γ-mean, geometric median and marginal median in the 2nd row of Figure 2 look collapsed together. To have a better view of these three bias curves, zoomed-in views are provided in the 3rd row of Figure 2, and we can clearly see that the γ-mean has the lowest bias values.

#### 4.2.3. Scenario 3

We conduct further experiments on γ-mean to see the effect of different γ values by setting γ=c/p with c=0.5,1,2,3,4, where α=0.1 and p∈[1,1000]. Results are presented in Figure 3. The squared bias is ignorable, indicating that the γ-mean is quite robust with different γ-values. The main source of MSE comes from the variance. Larger γ values lead to larger stochastic variances, indicating that larger γ values result in lower estimation efficiency.

#### 4.2.4. Scenario 4

For p∈[100,1000] and under the Gaussian case, the MSE, bias and variance curves for the γ-mean and simple γ-mean are collapsed together (Figure 4, left panel). For p∈[100,1000] and under the *t*-distribution, these curves are nearly indistinguishable (Figure 4, right panel). When *p* is small (p≪100), the MSE values of the simple γ-mean are significantly larger than those of γ-mean. However, we did not display the results for p<100, since the relative large MSE values for small *p* will make the MSE curves of γ-mean and simple γ-mean look lying on the *x*-axis for p≥100.

## 5. Real Data Examples

### 5.1. Datasets

MNIST [19]. The MNIST database of handwritten digits has a training set of 60,000 examples, and a test set of 10,000 examples. The digits have been size-normalized and centered in a fixed-size, 28×28 grayscale, images.Fashion MNIST [20]. Fashion-MNIST is a dataset of Zalando’s article images consisting of a training set of 60,000 examples and a test set of 10,000 examples. Each example is a 28×28 grayscale image, associated with a label from 10 types of clothing, such as shoes, t-shirts, dresses, sandals, sneakers and more.Chest X-ray images (pneumonia) [21]. The dataset contains 5856 X-ray images and 2 classes (pneumonia and normal). The 5,856 images consist of 5232 training images (which we further split into 90% for model training and 10% for model validation to implement early stopping) and 624 testing images. Chest X-ray images (anterior-posterior) were selected from retrospective cohorts of pediatric patients of one to five years old from Guangzhou Women and Children’s Medical Center, Guangzhou.

### 5.2. Experimental Setting

All the data examples were run on an Nvidia DGX A100 server. We used one A100 GPU card, 64 G RAM, and 16 cores of AMD-EPYC-7742 CPUs. The experimental setting is described below.

For MNIST and fashion MNIST, we set m=20 and the number of Byzantine clients is two. For chest X-ray images, we set m=9 and the number of Byzantine clients is one. Byzantine clients return random values from Gaussian (5, 1).γ is set to 0.5. This setting is different from the setting γ=c/p in simulation. The main reason is that there is a certain complicated relationship between the γ value and the neural network model adopted, such as the dimensionality, gradient size, learning rate, etc. We have not yet fully understood this relationship, which might govern the selection of γ. We will leave it as a future study.We set β=0.1 for the trimmed mean.To allow for the imbalanced size of clients, we obtain the sample size of each client by sampling from the following steps [11].Sampling a vector a from $\mathrm{Lognormal}(1.5,3.45^{2})$.Sampling η from Dirichlet(a). The sum of vector η will be 1 due to the property of Dirichlet distribution.Obtain the sample sizes of clients from multinomial $\left(n-m\times n_{o},\eta \right)$, where *n* is total sample size and $n_{o}$ is the minimum sample size guaranteed for each client. We set $n_{o}=512$.

In each round of parameter update, the server broadcasts the current parameter values to *m* clients. Each normal client computes the local gradient and local parameter update, and then returns the change of parameters after iterating *k* epochs using local data (k=1 for MNIST and fashion MNIST, and k=20 for chest X-ray images). Byzantine clients return random values from a Gaussian distribution N(5,1). Some further implementation details are given below.

We run 1000 rounds of FL for MNIST and fashion MNIST, and 100 rounds for chest X-ray images.We apply stochastic gradient descent (SGD) with a cosine decay learning rate (decay over rounds), where the decay step is 1000 for MNIST and fashion MNIST, and 100 for chest X-ray images. In each epoch of local clients on MNIST and fashion MNIST, the SGD will go through only 10% of local data to save computing time. This implementation leads to some fluctuations in the early stage of training but the training process will be much faster than going through all local data.The initial learning rates are 0.1, 0.5, and $10^{-4}$ for MNIST, fashion MNIST, and chest X-ray images, respectively. In each epoch in local iterates, $10^{-5}$ is set as the decay constant.We apply gradient clipping to avoid exploding gradients on MNIST and fashion MNIST. If the 2-norm of aggregated gradient is larger than 1, the vector will be scaled to a new vector with norm 1.To handle the imbalanced class size, we use weighted cross-entropy as loss function in the chest X-ray example (pneumonia: 0.35, normal: 1.0), where the chest X-ray training dataset contains 3883 pneumonia cases and 1349 normal cases. In addition to the classification accuracy, we also use ‘accuracy’, ‘sensitivity’ (also known as ‘recall’ and ‘true positive rate’) and ‘precision’ as our evaluation metrics. In particular, correctly predicting pneumonia is more important than predicting the normal case.

### 5.3. Models

We use a simple CNN model (Figure A1 in Appendix A) for MNIST and fashion MNIST. For chest X-ray images (pneumonia), we use a pretrained ResNet50 (Figure A2 in Appendix A) with input image size 150×150×3. We further connect the model to 3 dense-block layers.

### 5.4. Results

Results for the trimmed mean are not reported here, as they do not perform well due to its reliance on the specification of β and its non-robustness to non-symmetric outliers (note that the trimming procedure is symmetric in two tails of each coordinate).

#### 5.4.1. MNIST

Figure 5 shows the results of FL using different aggregators. In the case of no attack, the simple γ-mean performs as good as the mean-aggregator with 96% testing data classification accuracy. In the Byzantine case, simple γ-mean performs the best with testing accuracy 96% followed by the two median-based methods, geometric median and marginal median, while the mean-aggregator has failed. In addition, the marginal median has some fluctuations in the early stage of training.

#### 5.4.2. Fashion MNIST

Figure 6 shows the results of fashion MNIST. The conclusion is similar to MNIST, that the simple γ-mean performs the best. Note that the fluctuations shown in the training process of marginal median look worse than those shown in Figure 5. Although we have applied gradient clipping, still there are some sudden surges in loss function values and they lead to sudden drops in classification accuracy.

#### 5.4.3. Chest X-ray Images (Pneumonia)

We train the model on the training dataset without any partitions (i.e., single machine with full training data) as a baseline. In this single machine case, the maximum number of training epochs is 1000 and the training process will stop early if the evaluation metric (loss + accuracy + precision + sensitivity on validation data) does not get better in 10 consecutive epochs. For the FL case, the number of FL rounds is 100 and the maximum training epochs of each client are 20 at every FL round. At each round, the clients will stop early if the evaluation metric (loss + accuracy + precision + sensitivity on validation data) does not get better in 5 consecutive epochs. The simple γ-mean algorithm has adopted the marginal median as initial. All other settings are the same as those listed in Section 5.2. The results are shown in Figure 7.

We further report the following testing data results in Table 1: true positive (TP, which is predicted positive and actually positive), true negative (TN, which is predicted negative and actually negative), false positive (FP, which is predicted positive but actually negative), false negative (FN, which is predicted negative but actually positive), precision (Prec, which is $\frac{\mathrm{TP}}{\mathrm{TP}+\mathrm{FP}}$), sensitivity (Sens, which is $\frac{\mathrm{TP}}{\mathrm{TP}+\mathrm{FN}}$ as well as one minus type-II error) and classification accuracy (Acc). The aggregators by mean and geometric median cannot tolerate the Byzantine (Byz) attack and both methods were crushed during model training. Thus, there are no results reported for these two methods.

## 6. Concluding Remarks

Our major contribution in this article is to propose the γ-mean aggregation in federated learning, which is robust against outliers and Byzantine attacks. We have provided some theoretical discussions on influence functions and carried out numerical studies to justify our proposal. In both simulation and real data experiments, γ-mean based aggregation in general outperforms the existing robust aggregators such as marginal median, geometric median and trimmed mean. In addition, the simple γ-mean does not require complex computation. It can be easily computed by fixed point iteration and works well for extremely high dimensional cases such as in a deep neural network model. The γ-mean (as well as the simple γ-mean) takes a form of weighted average, where γ controls the trade-off between robustness and estimation efficiency. The selection of γ value is important, and a data-driven selection procedure is needed, which will be pursued in our future study.

In our numerical examples (simulation and real data), we have tested γ-mean’s ability to guard against the Byzantine attacks. However, when it comes to federated learning in real-world applications, adversarial attacks are also an important issue for model robustness. These attacks create adversarial fake examples, which even humans might not be able to distinguish from genuine ones. Moreover, adversarial examples aim to confuse the model and result in wrong prediction. How to modify the baseline γ-mean algorithm for adversarial attacks will be an interesting and yet challenging problem. We will defer it to a future study.

## Figures and Tables

**Figure 1 entropy-24-00686-f001:**
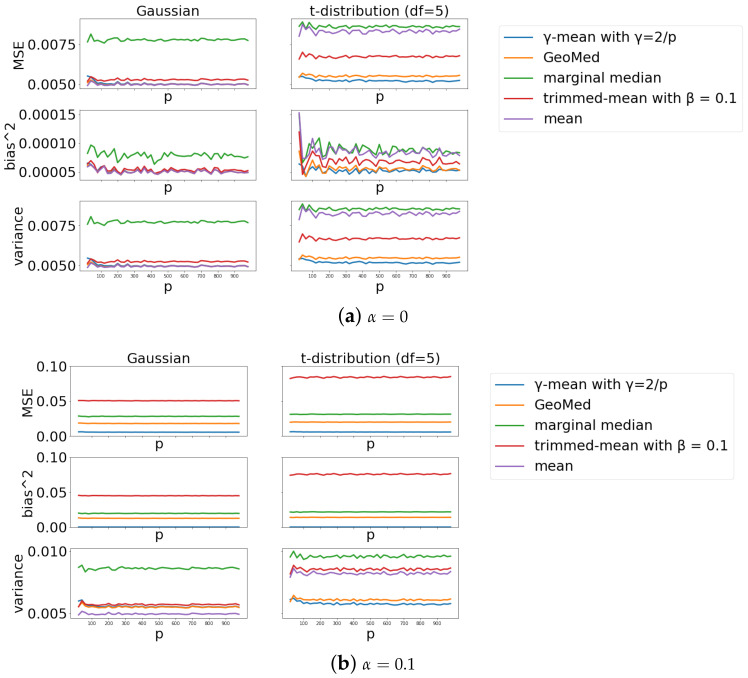
Comparison of different aggregators across different dimensions *p*. (**a**) Case α=0 (no Byzantine client). (**b**) Case α=0.1 (10% Byzantine clients).

**Figure 2 entropy-24-00686-f002:**
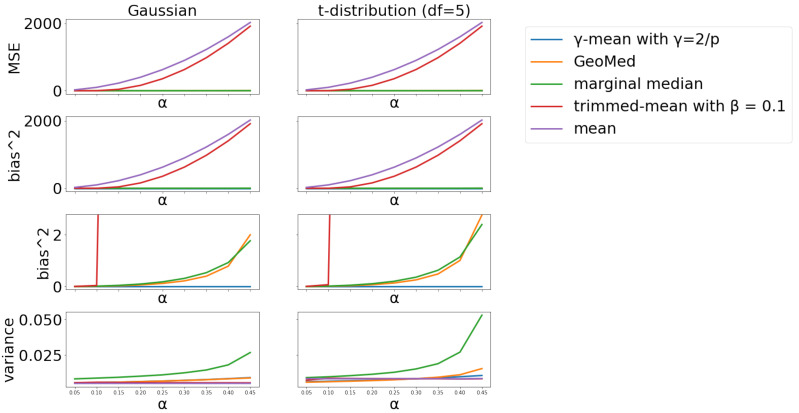
Comparison of different aggregators across different α values with fixed p=1000. The two subplots in the 3rd row are the zoomed-in views of subplots in the 2nd row.

**Figure 3 entropy-24-00686-f003:**
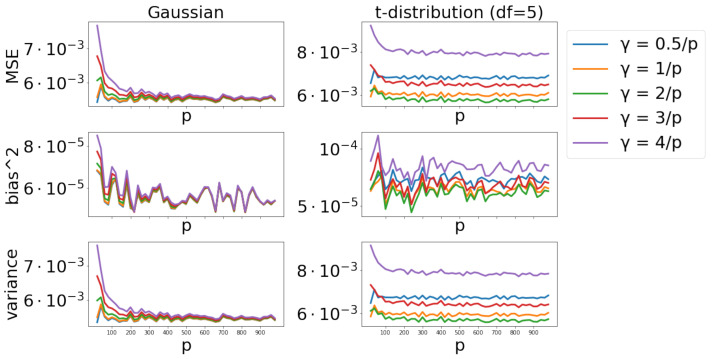
Effect of different γ values across different dimensions p∈[1,1000] with α=0.1.

**Figure 4 entropy-24-00686-f004:**
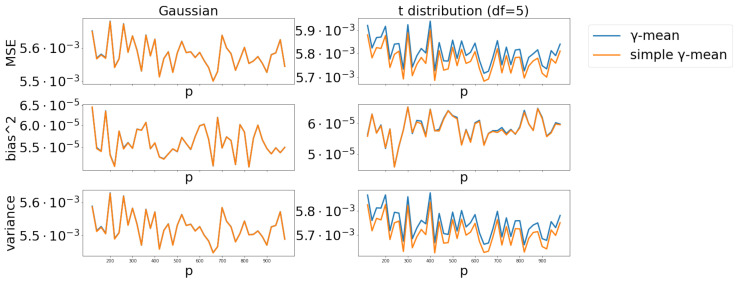
Comparison of γ-mean versus simple γ-mean with α=0.1 and p∈[100,1000].

**Figure 5 entropy-24-00686-f005:**
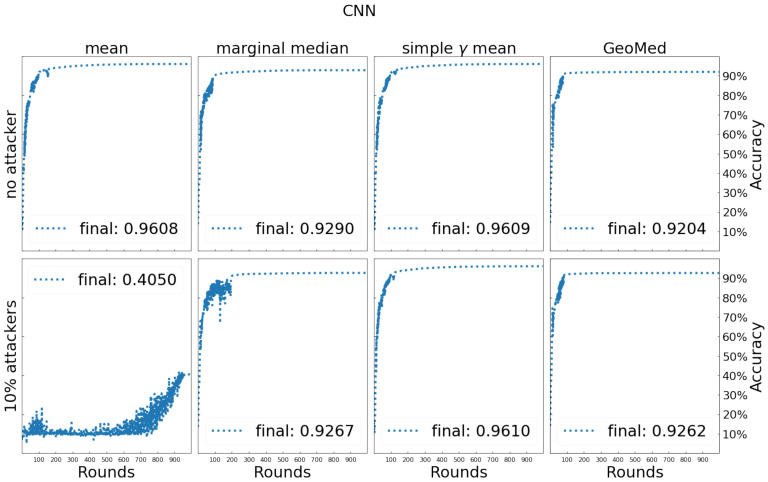
Testing process and comparison of testing accuracy for different aggregators on MNIST.

**Figure 6 entropy-24-00686-f006:**
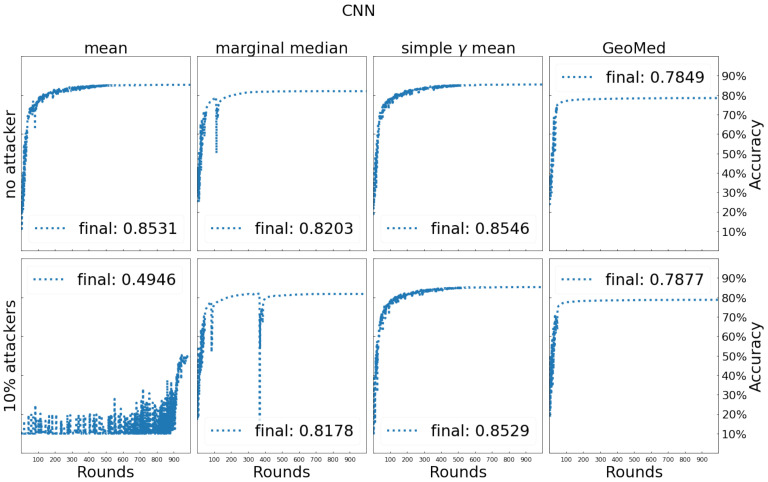
Testing process and comparison of testing accuracy for different aggregators on fashion MNIST.

**Figure 7 entropy-24-00686-f007:**
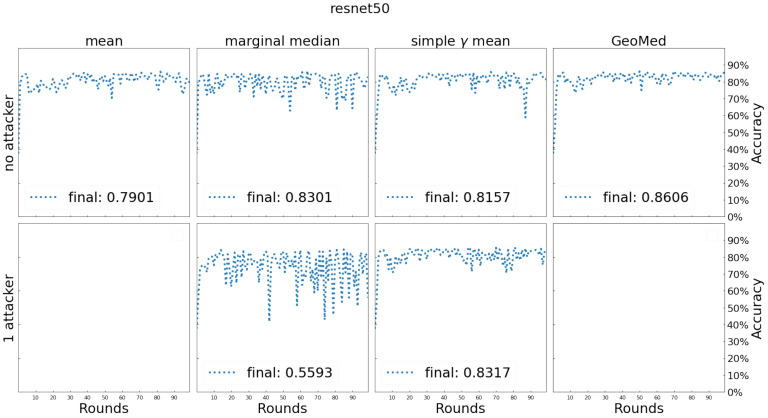
Testing process and comparison of testing accuracy for different aggregators on chest x ray. The aggregators by mean and geometric median cannot tolerate the Byzantine attack and both methods were crushed during model training. Thus, there are no results reported for these two methods.

**Table 1 entropy-24-00686-t001:** Pneumonia prediction on test data.

Byz	Aggregator	TN	FN	FP	TP	Prec	Sens (Type II Error)	Acc
	single machine	156	23	78	367	0.8247	0.9410 (0.0590)	0.8381
No	mean	212	103	22	287	0.9288	0.7359 (0.2661)	0.7997
	marginal median	190	63	44	327	0.8814	0.8385 (0.1615)	0.8285
	simple γ-mean	126	8	108	382	0.7796	0.9795 (0.0205)	0.8141
	GeoMed	177	30	57	360	0.8633	0.9231 (0.0769)	0.8606
Yes	mean ${}^{†}$	–	–	–	–	–	–	–
	marginal median	228	271	6	119	0.9520	0.3051 (0.6949)	0.5561
	simple γ-mean	140	11	94	379	0.8013	0.9718 (0.0282)	0.8317
	GeoMed ${}^{†}$	–	–	–	–	–	–	–

† The symbol “–” indicates model crushed during training.

## Data Availability

Benchmark datasets are available from the following links: “MNIST” at http://yann.lecun.com/exdb/mnist/ accessed on 31 August 2021, “fashion MNIST” at https://research.zalando.com/project/fashion_mnist/fashion_mnist/ accessed on 31 August 2021, and “Chest X-ray images (pneumonia)” at https://www.kaggle.com/datasets/paultimothymooney/chest-xray-pneumonia accessed on 1 October 2021.

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
