# Peer review of "Robust Aggregation for Federated Learning by Minimum γ-Divergence Estimation"

_entropy, 2022, doi:10.3390/e24050686_

Round 1
Reviewer 1 Report
The research topic of federated learning is of high interest for different entities and devices to collaboratively enter a global model without sharing their data and security is of vital importance. In federated learning with a central server, an aggregation algorithm integrates the model information sent by local clients to update the parameters of a global model. Although the sample mean has been the simplest and most widely used aggregation method, but it is not robust enough for data with outliers or under the Byzantine problem. As described in this paper there are some robust aggregation methods published, such as marginal median, geometric median and trimmed mean. In this paper, an alternative robust aggregation method, called γ-median, is presented, which is the estimation of the minimum divergence based on a robust density power divergence. The idea of this proposal pursues to mitigate the influence of Byzantine clients by assigning fewer weights. This weighting scheme is data-driven and controlled by the γ value. As an added value is the discussion on robustness from the influence function point of view along with the study based on the numerical results obtained.
The Robust aggregation by γ-mean algorithms presented are innovative and contribute to the state of the art. Moreover, the comparison with other aggregators is very interesting. The proposed scenarios help to understand the scope of the contribution. The article is well structured and presented.
Author Response
We thank you for supporting our ideas of using the γ-divergence for robust aggregation. Your positive comments are encouraging for us to continue on in this line of research for further development.
Reviewer 2 Report
The authors of this paper focused on federated learning and specifically on a new aggregation method, where the minimum divergence estimation is based on a robust density power divergence. To prove their point, the authors run simulations using the very popular MNIST dataset.
The paper is well written, however, there are issues, that the reviewer would like to raise with the authors.
Although the contributions are mentioned in the introduction, they can be summarized in bullet point format, which will make them more visible and easier to understand.
There are some elements of related work throughout the text but the section is missing. Although the area might not be thoroughly investigated yet, a separate section consisting of the background and the related work would greatly benefit the paper.
Although the simulation environment is detailed, the hardware setup of the testbed is missing.
Finally, there should be a discussion about the security of the introduced methodology and of course its susceptibility to adversarial attacks which have become more popular.
Round 2
Reviewer 2 Report
The authors have addressed all comments.